# Multi-Observer Study on the Assessment of Pediatric Gonadal Tumors Using Higher Harmonic Generation Microscopy as Compared to Conventional Histology

**DOI:** 10.3390/cancers17101636

**Published:** 2025-05-12

**Authors:** Sylvia Spies, Elina Nazarian, Felix Bremmer, Ivan A. Gonzalez, João Lobo, Miguel Reyes-Múgica, Eduardo Zambrano, Caroline C. C. Hulsker, Annelies M. C. Mavinkurve-Groothuis, Ronald R. de Krijger, Marie Louise Groot

**Affiliations:** 1LaserLab Amsterdam, Department of Physics, Faculty of Science, Vrije Universiteit Amsterdam, 1081 HZ Amsterdam, The Netherlandsm.l.groot@vu.nl (M.L.G.); 2Institute of Pathology, University Medical Center Göttingen, 37075 Göttingen, Germany; 3Department of Pathology and Laboratory Medicine, Indiana University School of Medicine, Indianapolis, IN 46202, USA; 4Department of Pathology, Portuguese Oncology Institute of Porto (IPO Porto), 4200-072 Porto, Portugal; 5Cancer Biology and Epigenetics Group, IPO Porto Research Center (GEBC CI-IPOP), CI-IPOP@RISE (Health Research Network), Portuguese Oncology Institute of Porto (IPO Porto), Porto Comprehensive Cancer Center Raquel Seruca (P.CCC), 4200-072 Porto, Portugal; 6 Department of Pathology and Molecular Immunology, ICBAS—School of Medicine and Biomedical Sciences, 4050-513 Porto, Portugal; 7Department of Pathology and Laboratory Medicine, University of Miami Miller School of Medicine, Miami, FL 33136, USA; 8Department of Pathology and Laboratory Medicine, Phoenix Children’s Hospital, Phoenix, AZ 85016, USA; 9University of Arizona College of Medicine, Phoenix, AZ 85004, USA; 10Creighton University School of Medicine, Phoenix, AZ 85012, USA; 11Princess Máxima Center for Pediatric Oncology, 3584 CS Utrecht, The Netherlands; 12Department of Pathology, University Medical Center Utrecht, 3584 CS Utrecht, The Netherlands

**Keywords:** pediatric gonadal tumors, germ cell tumors, nonlinear microscopy, third harmonic generation, second harmonic generation, two-photon excited autofluorescence, ovarian tumors, testicular tumors

## Abstract

Higher harmonic generation microscopy (HHGM) is an innovative imaging technique that enables rapid visualization of tissue without the need for preparation or staining. In this study, we demonstrate that HHGM provides comparable information about the gonadal architecture to conventional histology and pathologists can effectively discriminate between non-tumoral gonadal tissue and (pediatric) gonadal tumors. Therefore, HHGM could be used to rapidly assess the representativeness of a sample or biopsy for biobanking or other analyses that require a high tumor content.

## 1. Introduction

Pediatric gonadal tumors are rare tumors and germ cell tumors (GCTs) are the most frequent subgroup. GCTs are neoplasms that are derived from primordial germ cells. The subtypes of GCTs include embryonal carcinoma (EC), choriocarcinoma (CC), yolk sac tumor (YST), mature and immature teratoma, seminoma (where the term dysgerminoma is used for ovarian tumors and germinoma for extragonadal tumors), and mixed germ cell tumors (MGCT), consisting of several of the above histopathological components. Mature teratomas are benign tumors; all other histopathological GCT subtypes are considered malignant. Mature teratomas consist of well-differentiated tissue from one or more of the three germ cell layers: ectoderm, mesoderm, and endoderm. Immature teratomas are considered malignant and are graded based on the amount of primitive neuroepithelium. GCTs are heterogeneous and therefore require extensive sampling of the resected tumor to obtain an accurate diagnosis, as well as to prevent missing relevant components [1]. Other pediatric gonadal tumor subgroups are epithelial tumors, sex cord-stromal tumors and other types of tumors, such as rhabdomyosarcoma [2].

Pediatric ovarian tumors have an incidence of 2.6 cases per 100,000 children and are benign in 90% of cases [3]. GCTs are the most common ovarian tumors in females with a median age range from 16 to 20 years, but these tumors can also be found in children from 6 years old [4]. Ovarian teratomas show a peak around 12 years, while all malignant ovarian GCTs have a peak between 10 and 18 years [5]. Around one-third of GCTs are malignant, which makes them the majority of the malignant pediatric ovarian tumors, other malignant tumors include epithelial tumors and sex cord-stromal tumors [4,6]. Epithelial tumors are mainly serous cystadenomas and mucinous cystadenomas [2].

Pediatric testicular tumors have an incidence of 2 cases per 100,000 males [7]. More than 70% of the pre-pubertal testicular tumors are benign, while 75% of the post-pubertal tumors are malignant [8]. Pediatric testicular tumors are often GCTs or sex cord-stromal tumors. Testicular GCTs have a bimodal age distribution, the first peak is from 0 to 3 years and consists mostly of teratoma and YST, the second peak is from 12 to 18 years and consists predominantly of MGCT [5]. Testicular sex cord-stromal tumors are mostly Leydig cell tumors, followed by Sertoli cell tumors and juvenile granulosa cell tumors [9].

This study aims to assess whether a novel imaging technique, higher harmonic generation microscopy (HHG) microscopy, can provide rapid feedback on the representativeness of a GCT sample. We investigated the reproducible detection of normal gonadal tissue and pediatric gonadal tumors (comprising GCTs and other gonadal tumors) on HHG microscopy, with a group of expert pathologists evaluating the images. HHG microscopy is a high-resolution imaging technique that allows for the visualization of fresh, unprocessed tissue in a matter of minutes. It combines third harmonic generation (THG), second harmonic generation (SHG), and two-photon excited autofluorescence (2PEF) to visualize cellular structures and collagen and endogenous fluorophores, such as elastin, respectively. Previous studies demonstrate that HHG microscopy provides high-resolution images that provide architectural information of various types of tissue, including brain, breast, lung, and thyroid [10,11,12,13], and that artificial intelligence algorithms could provide assessment of malignancy [14,15].

The combination of SHG and 2PEF has been used to image pediatric tumors, but these studies used formalin-fixed and paraffin-embedded tissue [16,17]. In contrast to these studies, we include THG in our approach to visualize cells and cell nuclei, in addition to SHG and 2PEF, enhancing the overall information content and improving the resemblance to histological analysis. Moreover, our use of a transportable HHG microscope and fresh, unprocessed tissue underscores the potential of this technique for real-time imaging in the operating room.

## 2. Materials and Methods

Twenty-eight freshly excised tissue samples from twenty-two pediatric (0–18 years) patients treated or suspected for a GCT or another gonadal tumor in the Princess Máxima Center were imaged with the HHG microscope. After HHGM imaging, the samples were processed according to standard histopathological protocols. The HHGM images were compared with hematoxylin and eosin (H&E)-stained sections of the same tissue fragment. A multi-observer study was performed with five experienced pediatric germ cell tumor pathologists of the Malignant Germ Cell International Consortium (MaGIC).

### 2.1. Ethics Statement

All material was provided by patients from the Princess Máxima Center for pediatric oncology. The research was approved by the Biobank and Data Access Committee of the Princess Máxima Center for pediatric oncology (PMCLAB2022.297). All included patients underwent treatment at the Princess Máxima Center for pediatric oncology and provided written informed consent for participation in the biobank (International Clinical Trials Registry Platform: NL7744; https://onderzoekmetmensen.nl/en/trial/21619, accessed on 23 July 2024). This research project followed the Netherlands Code of Conduct for Research Integrity and the Declaration of Helsinki. The results of this research were not used for patient diagnosis or treatment.

### 2.2. Sample Handling

Tissue samples were transported from the operating room to the laboratory for pediatric oncology of the Princess Máxima Center in Utrecht, The Netherlands. One of the pathologists, who did not participate in the HHGM assessment, selected (vital) tumor and normal samples for HHGM imaging, based on macroscopic evaluation. There was no case selection, only a sample selection from the surgical specimen to obtain (suspected) tumor and, if possible, normal tissue samples. The samples, maximally measuring 10 mm × 10 mm × 5 mm, were placed in a sample holder (µ-dish 35 mm, high glass bottom, ibidi GmbH, Gräfelfing, Germany), with the same orientation as required for histology. HHGM imaging was performed on the bottom layer of the sample, around 20 microns from the glass interface. After imaging, the sample was processed following standard histopathology procedures. Hematoxylin and eosin (H&E) sections were made from the same cutting plane as the HHGM image to have the best correspondence. The gold standard diagnosis was considered as the diagnosis made by the pathologists of the Princess Máxima Center, based on histology of the entire tumor, and additional techniques such as immunohistochemistry and molecular techniques. The HHGM images were not involved in the diagnosis.

### 2.3. Image Acquisition

The set-up is shown in Figure 1A and B. A transportable HHG microscope (co-developed with Flash Pathology B.V. Amsterdam, The Netherlands) was used to acquire the HHGM images. A brief explanation on the physical mechanism of HHGM is given in the Appendix B. The HHG microscope was similar as described previously [12,13], with minor modifications. A sub80-fs laser source, centered at 1060 nm (Biolit 2 with precompensation, Litilit, Vilnius, Lithuania) was used to generate the nonlinear signals. The microscope was equipped with an acousto-optic modulator (AOM) to select bunches of 5–10 pulses at a repetition rate of 1 MHz out of the 15 MHz pulse train, to achieve a low average power of 5 mW, with a pulse peak energy of 5 nJ. This ensured a sufficient peak power to generate the nonlinear optical signals while, at the same time, a low average power avoids damaging (heating) of the tissue. The laser beam was focused using an oil-immersion microscope objective with high numerical aperture (40×/1.3NA, Nikon, Tokyo, Japan) to obtain a sub-micrometer focus of 0.4 × 0.4 × 2.4 μm^3^ [12]. The sample dish was placed in a sample holder above the objective. Analogue photo-multiplier tubes (H10721-210 and H10721-20, Hamamatsu Photonics, Hamamatsu, Japan) collected the third harmonic generation (THG), second harmonic generation (SHG), and two-photon excited autofluorescence (2PEF) signals. Dichroic mirrors and interference filters were used to separate the three signals, resulting in the following detection bandwidths: THG 335–376 nm, SHG 505-545 nm, and 2PEF 573–642 nm. Two-dimensional images were acquired by bidirectional raster scanning of the galvanometer mirrors. Larger 2D images were created as a mosaic of smaller images by moving the sample in x- and y-direction (XYZ stage, Applied Scientific Instrumentation, Eugene, OR, USA) after each small 2D image. Mosaic scanning of preset dimensions was automatically performed with a LabVIEW program (Flash Pathology B.V.). In addition, a multi-axis stage controller (Applied Scientific Instrumentation) with a joystick (x, y-direction) and control wheel (z-direction) could be used to inspect the sample manually.

Images were acquired with a LabVIEW program (Flash Pathology B.V.), which had preset scanning programs with different field of views and sampling densities, and therefore different acquisition times. The characteristics of the scanning programs are listed in Figure 1C. The Fast Overview scan was used to make an overview image of the entire sample. The high-quality scan was used to scan selected regions of interest (usually regions of 2 mm × 2mm), often the cellular regions. Imaging time varied around 20 min to one hour and depended mostly on the surface area of the sample and the number of high-quality scans taken. The raw data contain signal intensities between 0 and 10,000 and were converted to 24-bit RGB images (8-bit for each color, i.e., intensities between 0 and 255). The signal intensities were scaled with a gamma correction in the LabVIEW program: γ = 0.5 for THG and SHG and γ = 0.7 for 2PEF signal. The upper limit of the display range for each signal was manually determined to make sure that the cells and structures were well visible, without creating oversaturated images. Each signal is visualized in a different color: THG in green, SHG in red, and 2PEF in blue. The images are a mixture of these colors. For example, collagen can generate SHG and THG, which results in an orange or yellow appearance in the HHGM images.

### 2.4. Pathologists Assessment

#### 2.4.1. Slide Score

To investigate the ability of the pathologists in interpretation of the HHGM images, a study was set up using Slide Score (https://www.slidescore.com, Amsterdam, The Netherlands), an online pathology platform which integrates a pathology viewer with a question sheet. The pathology viewer could be used to zoom in on the images and navigate, to analyze the large images in detail. In addition, pathologists were able to enhance the colors of the images or to (de)select different color channels.

#### 2.4.2. Training of the Pathologists

The pathologists received a document, which explained the technique and the meaning of the different colors. The pathologists were allowed to use the training document during the assessment. To prevent any bias, only general examples were given (i.e., adipose tissue, cartilage, muscle, skin, immune cells, artifacts) and small (400 µm × 400 µm) close-ups of some histopathological features: seminiferous tubules, blood vessel, embryonal carcinoma, seminoma/dysgerminoma, and yolk sac tumor. These images were rotated and/or mirrored and changed in color intensity so that they could not be recognized in the assessment. Furthermore, the Slide Score study started with a training set, in which HHGM images and corresponding histology images from two excluded gonadal samples (non-relevant tumors) were shown simultaneously. This training case was used to practice with the pathology viewer and the question sheet, and to see how the HHGM images correspond with the histology images.

#### 2.4.3. Study Design

To mimic the normal workflow of the pathologist, limited case information was provided (age, gender, and organ) and questions were based on how histology sections are usually evaluated (Appendix A). The order of the cases was the same for each pathologist, so that they all had the same learning effect (Appendix A). For each case, pathologists first assessed the HHGM part (overview image of the entire sample and the high-quality close-ups). After the HHGM part, the pathologists received the corresponding H&E section with the same question sheet and could not see nor change their assessment of the HHGM part.

#### 2.4.4. Data Analysis

The consensus, determined by the majority opinion (≥3 of the pathologists), was used to calculate the diagnostic characteristics (sensitivity, specificity, positive predictive value, negative predictive value, and accuracy). Confidence intervals of 95% were calculated similar to MedCalc [18]. Clopper–Pearson confidence intervals were used for sensitivity, specificity, and accuracy. Standard logit confidence intervals given by Mercaldo et al. [19] were used for PPV and NPV, except for the values that reached 0% or 100%: these were calculated with Clopper–Pearson confidence intervals.

## 3. Results

Twenty-eight samples were obtained from twenty-two patients with a GCT or other gonadal tumor (Appendix A). The complete results of the pathologist’s assessment can be found in the Appendix A (S3: normal testis, S4: nontumoral ovary, S5: teratomas, S6: non-teratoma GCTs, S7: non-GCTs). These tables also show the comments of the pathologists regarding the features that they could recognize in the images, such as seminiferous tubules and Leydig cells.

### 3.1. Normal Versus Tumor

Four samples were collected from normal testis tissue, determined based on macroscopic evaluation to be sufficiently far away from the tumor. However, it was not possible to completely exclude that these samples contained areas with the precursor of GCTs: germ cell neoplasia in situ (GCNIS). For the purpose of this study, we considered GCNIS as non-tumor tissue. Additionally, three samples were obtained from suspected ovarian lesions. However, the gold standard diagnosis indicated no malignancy, specifically a follicular cyst, normal ovarian tissue, and necrosis. These samples were also considered non-tumor tissue.

Compared to the gold standard diagnosis, HHGM achieved 75% (21/28), while H&E reached 89% (25/28), as shown in Table 1A,B, respectively. Notably, almost all tumor cases that were misclassified as non-tumor on HHGM or H&E images, were teratoma samples. Table 1C shows the accuracy of HHGM compared to H&E, instead of the ground truth, which yields an accuracy of 79% [59–92%].

### 3.2. Comparison with Histology

Figure 2 shows a comparison of HHGM and H&E for normal testis (Figure 2A,B) and some germ cell tumors (Figure 2C–G). The tubular architecture of normal testis is visible in the HHGM image, and the tunica albuginea can be recognized as a thick layer of connective tissue. The tumor cases show a loss of the tubular architecture. The HHGM images show architectural patterns very similar to the histology images.

### 3.3. Germ Cell Tumors: Teratomas

Most of the germ cell tumors in this study were teratomas (9/16). As shown in Table 2, 44% (4/9) of the teratoma samples were correctly recognized as tumor on HHG, and 75% (3/4) of those were correctly diagnosed as teratoma. On histology, 67% (6/9) of the samples were correctly recognized as tumor and 100% (6/6) of those were classified as teratoma. Cases that were not interpreted as tumor on HHGM and H&E were mainly cases that consisted predominantly of connective tissue and/or muscle. The results per pathologist are shown in the Appendix A, which shows that the results vary between pathologists, but two very heterogeneous cases were diagnosed as teratoma by all five pathologists on both HHGM and H&E. Figure 3 shows a comparison of HHGM and H&E for some teratoma examples.

### 3.4. Other Germ Cell Tumors

In addition to teratomas, this study included seven other germ cell tumors: one embryonal carcinoma, two seminomas/dysgerminomas, one yolk sac tumor, and three mixed germ cell tumors. As shown in Table 3, all seven non-teratoma cases were correctly classified by the pathologists as tumor on both the HHGM and the H&E images. The YST case was correctly diagnosed on HHGM, although for the other cases there was no clear consensus. The H&E consensus diagnoses were in correspondence with the gold standard diagnoses. The result for each pathologist can be found in the Appendix A, which also highlights that the H&E diagnosis was not always correct, and in some cases the pathologists made the correct diagnosis using HHG.

### 3.5. Non-Germ Cell Tumors

This study includes five non-germ cell tumors from the ovary and testis: one mucinous cystadenoma, one Leydig cell tumor, one Sertoli cell tumor, and two rhabdomyosarcomas. The results of these cases are presented in Table 4. Four of the five cases were classified as tumor on HHG; however, there was no clear consensus on the tumor type. The pathologists’ consensus diagnosis for the H&E images correctly classified and diagnosed all cases. The results for each pathologist are provided in the Appendix A, which also highlights that the diagnosis on H&E was not always correct. In addition, some pathologists were able to identify features on the HHGM images that were characteristic of the diagnosis, such as ‘epithelial’ for the mucinous cystadenoma and ‘spindle cells’ for rhabdomyosarcoma. Figure 4 shows a comparison of HHGM and H&E for mucinous cystadenoma, rhabdomyosarcoma, and Leydig cell tumor. Notably, the mucinous cystadenoma shows how a single layer of epithelium can appear as a multi-layered structure on the HHGM images, depending on the sample’s orientation. The Leydig cell tumor example illustrates the importance of analyzing individual color channels.

## 4. Discussion

In this study, we investigated the use of HHGM for the recognition of pediatric normal gonadal tissue and gonadal tumors by a group of expert pathologists. Our findings reveal that pathologists could recognize the tissue architecture in the HHGM images. The diagnostic accuracy for distinguishing tumor from non-tumor was 75% (21/28) for HHGM, compared to 89% (25/28) for H&E-stained images.

Our study demonstrates that HHGM corresponds well with histology in terms of tissue architecture, and pathologists were able to differentiate between non-tumor gonadal tissue and tumor tissue. These findings suggest that HHGM could be similarly effective for other organs with distinct differences between normal and tumor architecture.

Given its speed, HHGM has potential for intraoperative applications. However, intraoperative feedback for pediatric gonadal tumors is typically not required, except when there is uncertainty in the diagnosis of GCT versus epithelial tumor, or to check whether peritoneal deposits are malignant. More practically, HHGM could be used to assess sample representativity, as a non-tissue consuming alternative for frozen sections, for example, to select material for biobanking, where the histological confirmation of tumor presence is important before pursuing expensive studies [20].

HHGM may have further clinical relevance for tumors where organ-sparing surgeries are more common, and where intraoperative frozen sections are used to guide the extent of resection. In our study, we utilized a transportable HHG-microscope and unprocessed tissue, demonstrating the feasibility of applying this technique directly in the operating room for immediate analysis. Furthermore, HHGM can be integrated into a needle or endoscope [10], enabling in vivo applications such as the assessment of tumor margins during surgery.

The results in the Appendix A show some diagnostic challenges in conventional histology as well. For the non-tumor cases, there was some interobserver variability in the classification of normal versus GCNIS and normal/reactive versus various benign tumors. Seminoma was classified by two pathologists as embryonal carcinoma, and Leydig cell tumor was classified as yolk sac tumor by one pathologist. These results show that even on histology there is variation between pathologists, and therefore it is difficult to determine the diagnostic accuracy.

This study has a few limitations. Since HHGM was performed on freshly excised tissue fragments, only a limited number of samples were available and most of them were composed of tumor tissue. Some tumor types only appeared once in this study or not at all. Therefore, we have too little data to draw conclusions about the diagnostic accuracy of HHGM for tumor diagnosis. To assess the diagnostic value of HHGM, other diagnosis than tumors should also be included. With regard to samples, we were limited to the resection specimens in the Princess Maxima Center, which is a pediatric oncology center, with only occasional non-tumor cases. In addition, samples were freshly selected based on macroscopic examination and thus were not always representative of tumor. This is particularly evident from the fact that even on histology, some tumor samples were not recognized as tumor, and some nontumor samples were diagnosed on histology as GCNIS, disorders of sexual development, fibroma or other (benign) tumors, with significant interobserver variability. The heterogeneity of the teratoma samples had a significant impact on their recognition. All pathologists correctly diagnosed the two most heterogeneous teratoma samples on HHGM, as well as H&E. However, samples predominantly composed of connective tissue and muscle were not interpreted as tumor, even on histology. Furthermore, the HHGM and H&E images were not one-to-one comparable. The samples were processed for histology after HHGM imaging and sections were made from the same cutting plane, but not from the same level. In one case (Sertoli cell tumor), the tumor could have been in a deeper level than the HHGM image. In another case (teratoma), the H&E section could not be analyzed. These samples were also smaller (around 5 × 5 × 5 mm^3^) than most samples (around 10 × 10 × 5 mm^3^), which could make it difficult to maintain the same orientation for HHGM and H&E. To obtain H&E sections that are better comparable to the HHG images, the sample should have a flat surface so that it is easy to maintain its orientation, and the section should be cut from a superficial layer.

To mimic the normal workflow of a pathologist, the patient’s age, gender, and the involved organ were provided, which may have introduced a bias in the tumor diagnosis. However, non-tumor tissue was also included, requiring pathologists to differentiate between nontumor and tumor tissue. Additionally, the interobserver variability was high, for both HHGM and H&E, complicating results’ interpretation and subsequent calculations. As a result, the diagnostic characteristics were determined based on the consensus opinion, defined as the majority opinion (≥3 of the pathologists).

Pathologists had limited training in interpreting the HHGM images, which contributed to some diagnostic challenges. Most false positives in HHGM occurred with normal reference testis samples, which were mistakenly interpreted as seminomas. One pathologist noted that the seminiferous tubules appeared more packed in the HHGM images, raising suspicion for (intratubular) seminoma. Additionally, certain tissue types were not recognized in teratoma cases. For example, intestinal epithelium and neuropil were not recognized, even though these tissue types were distinct from the surrounding collagen. On HHGM images, neuropil and epithelium both appear green, while collagen appears red. This example highlights that interpreting HHGM images is not always intuitive, and additional training is required to improve recognition of these features.

A similar challenge was observed in detecting lymph node metastasis. A lymph node with dysgerminoma metastasis showed a loss of normal architecture, and with tumor cells that were significantly larger than the lymphocytes and had a prominent nucleolus. However, this tumor was correctly diagnosed on HHGM images by only one pathologist. Therefore, more training on interpreting HHGM images of normal and tumor tissue is necessary to enhance the diagnostic accuracy. This training would consist of a set of HHGM and corresponding H&E images of different types of normal tissue and tumors. This would show how HHGM images correspond to H&E images, and which architectural and cellular features can be recognized in HHGM images.

Some HHGM images contained artifacts, such as bright speckles caused by Indian ink that was used to indicate surgical margins, or high autofluorescence and tissue deformation from cauterization during surgery. Other artifacts arose from the set-up, including air bubbles in the immersion oil or beneath the sample, which create shadow-like effects on the images. These artifacts usually only affect part of the image (usually shadows/dark areas of less than 100 × 100 micron) and therefore have limited influence on the diagnostic accuracy. Similar to histology, it is important to be aware that these artifacts exist and know how to deal with them. Most artifacts are comparable with the artifacts in conventional histology, for example, cauterization and crush artifacts create tissue deformation, and air bubbles or foreign objects can be present in the image. On the other hand, HHG does not have artifacts that are often seen in histology, such as tears, folds/wrinkles, or out-of-focus areas.

Furthermore, color contrast may affect the interpretation of the image. Specifically, the 2PEF and SHG signals exhibited significant variations in intensity, even within the same sample. The 2PEF signal was often weak, except in cases with cauterization artefact. These low-intensity features are difficult to visualize, also because the colors overlap, and because blue colors are not well perceived by the eye. Increasing the intensity of lower signals often results in oversaturation of higher intensity structures. To maintain consistency and preserve relative signal differences, similar values were used to generate all images. To compensate for this, the pathology viewer in Slide Score allowed for the adjustment of individual channels and intensities, enabling weaker features to be analyzed more effectively. Additionally, since the images were analyzed digitally, differences in the screens used by the pathologists could have led to variations in how the images were perceived.

In this study, imaging time ranged from 30 min to a maximum of one hour, primarily depending on the size of the sample and the number of high-quality scans. A fast overview image took 0.9 s per tile (300 × 300 µm^2^), corresponding to 10 s per mm^2^, while a high-quality scan required 2.4 s per tile (200 × 200 µm^2^), or 1 min per mm^2^. To ensure optimal comparison with histology, the entire bottom surface area of the sample was scanned using the fast overview mode, and around 3 high-quality scans (usually 4 mm^2^ each) were made. For samples of around 5 mm x 5 mm were used, the fast overview scan would take approximately 4 min. This fast overview scan was sufficient for identifying cellular areas and tissue architecture. Additionally, during scanning the software immediately displayed the imaged tile, and it is possible to manually inspect the sample using a joystick, enabling a quick inspection of a large sample.

## 5. Conclusions

Pathologists were able to identify the architectural differences between normal gonadal tissue and tumor tissue using HHGM images. The diagnostic accuracy of HHGM for classifying non-tumor versus tumor was 75% [55–89%], compared to 89% [72–98%] for H&E. Discrepancies were primarily observed in the teratoma cases for both HHGM and H&E, suggesting that sampling errors in heterogeneous tumors negatively impacted the study’s outcomes.

While we lacked sufficient data to accurately assess the diagnostic performance for specific tumor types, we found that all malignant tumors were correctly classified as tumors, and all false negatives were benign tumors. Additionally, HHGM offers advantages over frozen section analysis, including faster processing and the fact that it is non-tissue consuming. In conclusion, despite the limited sample size HHGM shows potential for intraoperative use, particularly for assessing sample representativity or identifying malignancy.

## Figures and Tables

**Figure 1 cancers-17-01636-f001:**
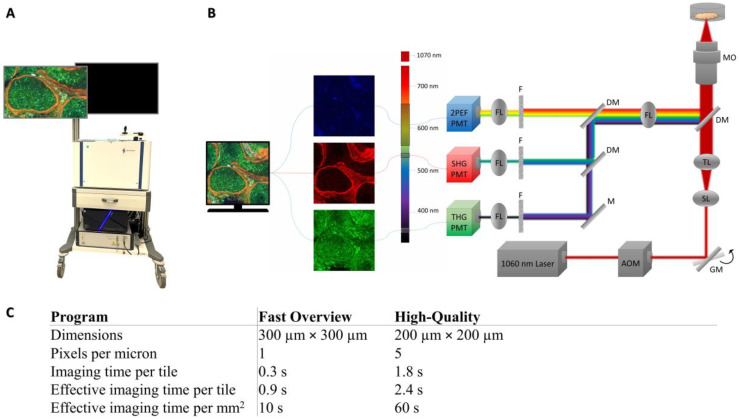
Image of the transportable HHG microscope (**A**) and schematic set-up (**B**) used in this research. The set-up includes a near-infrared laser (1060 nm), an acousto-optic modulator (AOM), galvanometer mirrors (GM), scan lens (SL), tube lens (TL), dichroic mirrors (DM), microscope objective (MO), focus lens (FL), mirror (M), filters (F), and photo-multiplier tubes (PMT). (**C**) Characteristics of the scanning programs. The effective imaging time also includes the movement of the XYZ-stage, which takes around 0.6 s per image.

**Figure 2 cancers-17-01636-f002:**
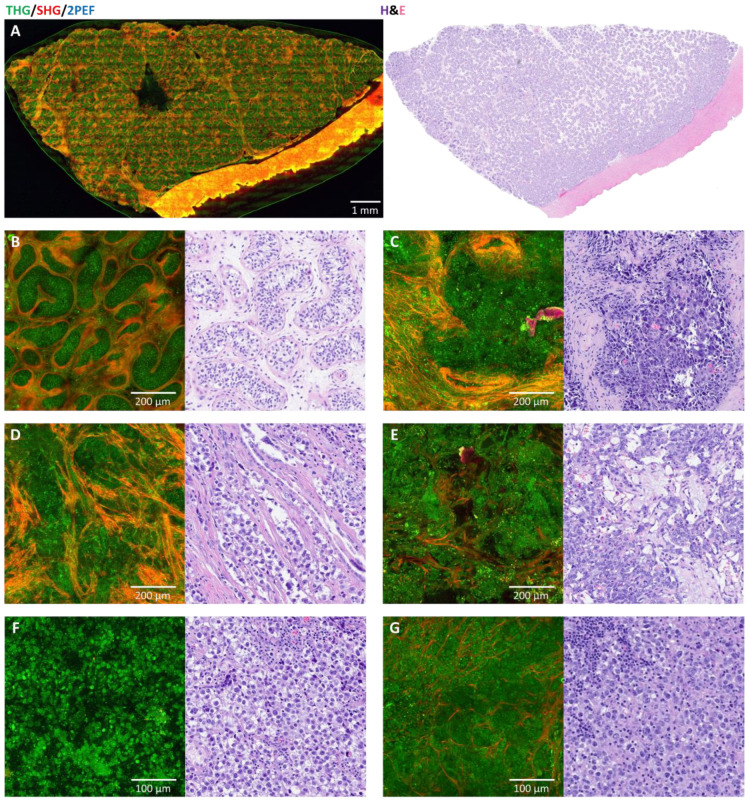
Comparison of HHGM and H&E for normal testis (**A**,**B**) and some germ cell tumors (**C**–**E**). (**A**) Normal prepuberal testis with a tubular architecture and a thick layer of connective tissue at the lower right border (tunica albuginea). (**B**) High-resolution close-up of the seminiferous tubules. (**C**) Embryonal carcinoma with a compact structure and indistinct cell borders. (**D**) Seminoma with sheet-like patterns of cells which are separated by fibrous septa. (**E**) Yolk sac tumor with a microcystic pattern. (**F**) Seminoma cells floating in the surrounding fluid of the sample are better distinguishable and show prominent nucleoli. (**G**) Dysgerminoma cells in a lymph node metastasis show that the tumor cells are larger than the lymphocytes and have sometimes multiple prominent nucleoli.

**Figure 3 cancers-17-01636-f003:**
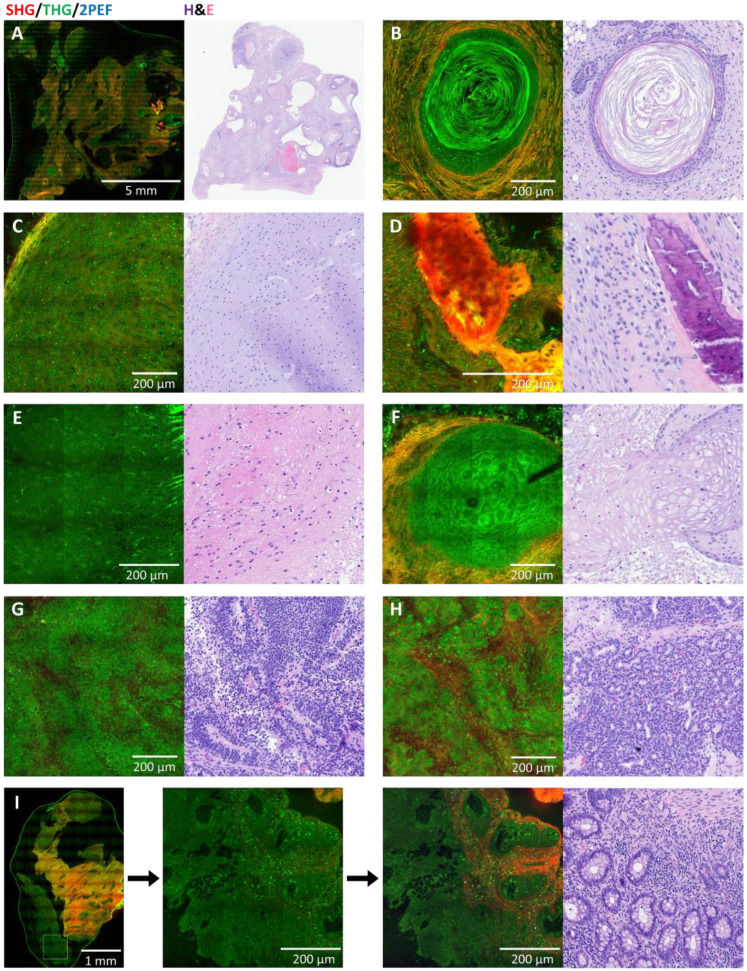
Comparison of HHGM and H&E for different elements in teratomas. (**A**) Overview image of a heterogeneous teratoma sample. (**B**–**H**) High-resolution examples of various elements found in teratoma samples: keratin cyst (**B**), cartilage (**C**), bone fragments (**D**), brain neuropil (**E**), squamous epithelium (**F**), and immature elements with primitive neuroepithelium (**G**,**H**), where (**H**) shows rosette-like structures. (**I**) Overview image of a teratoma sample with mainly collagen and muscle, and a border of intestinal epithelium at the lower left, the high-resolution close-up shows that increasing the SHG (red) intensity improves visualization of the crypts.

**Figure 4 cancers-17-01636-f004:**
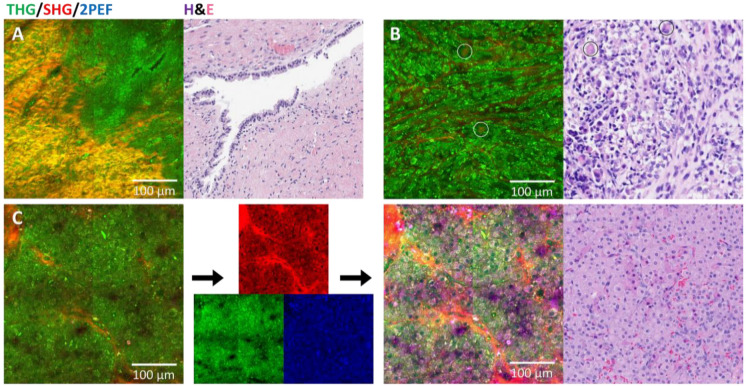
Comparison of HHGM and H&E for non-germ cell tumors. (**A**) Mucinous cystadenoma showing that a single layer of epithelium can have a multilayer appearance on the HHGM images. (**B**) Rhabdomyosarcoma, showing spindle cells and some cells that generate SHG within their cytoplasm (circled in white), which could be rhabdomyoblasts (circled in black). (**C**) Leydig cell tumor, showing that enhancement of the weak signals and inspecting individual channels sometimes improves the visualization. In this case, the enhanced red SHG channel had the clearest contrast of the cell nuclei (dark holes), which may also be caused by autofluorescence in the SHG detection wavelength.

**Table 1 cancers-17-01636-t001:** Diagnostic characteristics: (**A**) HHGM consensus compared with the gold standard diagnosis, (**B**) H&E consensus compared with the gold standard diagnosis, (**C**) HHGM consensus compared with H&E consensus. 95% confidence intervals are given in square brackets. Complete results per pathologist can be found in Appendix A.

(**A**)					
		Gold Standard Diagnosis			
		Tumor	Non-Tumor		
HHGM Consensus	Tumor	15	1	PPV = 94% [71–99%]	
	Non-Tumor	6 *	6	NPV = 50% [32–68%]	
		Sensitivity = 71% [48–89%]	Specificity = 86% [42–100%]	Accuracy = 75% [55–89%]	
(**B**)					
		Gold Standard Diagnosis			
		Tumor	Non-Tumor		
H&E Consensus	Tumor	18	0	PPV = 100% [81–100%]	
	Non-Tumor	3 **	7	NPV = 70% [45–87%]	
		Sensitivity = 86% [64–97%]	Specificity = 100% [59–100%]	Accuracy = 89% [72–98%]	
(**C**)					
		H&E Diagnosis			
		Tumor	Non-Tumor		
HHGM Consensus	Tumor	14	2	PPV = 88% [66–96%]	
	Non-Tumor	4	8	NPV = 67% [44–83%]	
		Sensitivity = 78% [52–94%]	Specificity = 80% [44–97%]	Accuracy = 79% [59–92%]	

* 5 cases were teratomas and the other case was probably imaged just next to the tumor. ** All 3 cases were teratomas and one of them was torn apart in the H&E.

**Table 2 cancers-17-01636-t002:** Contingency table of HHGM consensus versus H&E consensus of the nine teratoma cases. Complete results per pathologist can be found in Appendix A.

	H&E Consensus				
		Tumor, Teratoma	Tumor, Else	Non-Tumor	Total
HHGM Consensus	Tumor, teratoma	3	0	0	3
	Tumor, else	0	0	1 *	1
	Non-tumor	3 **	0	2 **	5
	Total	6	0	3	9

* H&E was torn apart. ** Cases with mostly connective tissue and/or muscle.

**Table 3 cancers-17-01636-t003:** HHGM versus H&E consensus of the seven other germ cell tumors. Complete results per pathologist can be found in Appendix A.

Gold Standard Diagnosis	HHGM Consensus	H&E Consensus
Embryonal carcinoma (testis)	Tumor, no consensus	Tumor, EC
Dysgerminoma metastasis (lymph node)	Tumor, no consensus	Tumor, dysgerminoma
Seminoma (testis)	Tumor, seminoma or EC	Tumor, seminoma
YST (testis)	Tumor, YST	Tumor, YST
MGCT (testis)	Tumor, no consensus	Tumor, MGCT: EC/YST
MGCT, predominantly teratoma (abdomen)	Tumor, no consensus	Tumor, teratoma
MGCT (testis)	Tumor, no consensus	Tumor, MGCT: EC/YST

**Table 4 cancers-17-01636-t004:** HHGM versus H&E consensus of the five non-germ cell tumor cases. Complete results per pathologist can be found in Appendix A.

Gold Standard Diagnosis	HHGM Consensus	H&E Consensus
Mucinous cystadenoma (ovary)	Tumor, no consensus	Tumor, (serous/mucinous) cystadenoma
Leydig cell tumor (testis)	Tumor, no consensus	Tumor, Leydig cell tumor
Sertoli cell tumor (testis)	Non-tumor *	Tumor, Sertoli cell tumor
Rhabdomyosarcoma (testis)	Tumor, no consensus	Tumor, Rhabdomyosarcoma
Rhabdomyosarcoma (testis)	Tumor, no consensus	Tumor, Rhabdomyosarcoma

* Tumor was probably deeper in the tissue than the HHGM image.

## Data Availability

The datasets generated during and/or analyzed during the current study are available from the corresponding author on reasonable request.

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
