# Peer review of "Multi-Observer Study on the Assessment of Pediatric Gonadal Tumors Using Higher Harmonic Generation Microscopy as Compared to Conventional Histology"

_cancers, 2025, doi:10.3390/cancers17101636_

Round 1

Reviewer 1 Report

Comments and Suggestions for Authors

Overview, strengths and limitations

This study investigates the application of higher harmonic generation microscopy (HHGM) in comparison with conventional histology (H&E) for evaluating paediatric gonadal tumours (germ cell tumours as well as non-germ cell tumours). The primary goal is to assess whether HHGM, by quickly visualizing fresh tissue without the need for processing or staining, can yield structural details like those obtained through standard histopathological techniques. Five experienced pathologists independently reviewed HHGM images alongside corresponding H&E sections, concentrating on diagnostic accuracy in distinguishing between tumour and non-tumour tissues, with special emphasis on germ cell tumours and challenging teratoma cases.

A total of 28 samples from 22 paediatric patients were analysed, including both tumour specimens and normal tissues (testis and ovary). HHGM images were acquired using both rapid overview and high-resolution scanning protocols. These were then directly compared with conventional H&E sections prepared from the same samples. The overall diagnostic accuracy was 75% for HHGM, compared to 89% for H&E and the most significant discrepancies were observed in teratoma cases, probably related to tissue heterogeneity and sampling variability.

The authors employed contingency tables and calculated confidence intervals to evaluate the data, demonstrating that while conventional histology outperforms HHGM, the latter shows promise as a rapid, tissue-sparing diagnostic method.

This study shows strengths and limitations.

HHGM offers a novel, interesting, method for immediate tissue assessment without extensive preparation, which is particularly useful for intraoperative decision-making and for selecting specimens for bio-banking.

The strengths are undoubtedly represented by the robust multi-observer assessment due to the involvement of five expert pathologists (which increases the reliability of the study by providing a detailed picture of the inter-observer variability); and by a direct comparative analysis which allows a head-to-head comparison with the gold-standard H&E method.

The limitations are represented by limited sample size and variability (only 28 cases and notable heterogeneity of germ cell tumours) which make the results not immediately generalizable; and the imaging artifacts (such as air bubbles and cauterization effects) and challenges in detecting low-intensity signals (notably the 2PEF component) indicate that further refinement of the imaging protocol is probably necessary.

Publication Recommendations

The manuscript fits well within the Special Issue “Digital Pathology System Enabling the Quality of Cancer Patient Care” and offers an innovative approach with potentially relevant implications in research and clinical practice.

Minor Revisions Recommended

  • Provide, if feasible, quantitative details on the impact of imaging artifacts on HHGM performance.
  • Expand the discussion on necessary training measures to enhance the consistency of HHGM image interpretation among pathologists (which could be especially useful for general pathologists).
  • Attempt to describe a strategy to address methodological issues, such as possible sectioning discrepancies, to improve the future applicability of the method.

Language

The manuscript is written in clear and technically appropriate English. Some sentences are a bit complex and could be simplified to improve overall clarity and readability.

Conclusion
In my opinion, this article makes a valuable contribution to digital pathology by introducing HHGM as a potential rapid diagnostic tool both in research and clinical practice. With further minor revisions addressing methodological issues and discussion, the manuscript appears well-suited for publication in Cancers within the specified Special Issue.

Reviewer 2 Report

Comments and Suggestions for Authors

The article is well-written and addresses an important topic—the potential use of  HHGM for the histopathological diagnosis of rare tumors, specifically pediatric germ cell tumors. The research is commendable for exploring innovative methods to improve diagnostic speed and efficiency, especially in cases where traditional histological analysis may be time-consuming or impractical.

The rarity of these tumors naturally limits the number of cases available for study, which is well acknowledged by the authors. The small sample size (28 samples from 22 patients) inevitably constrains the statistical power of the study, meaning that reported values for accuracy, sensitivity, and specificity must be interpreted with caution. This limitation is appropriately discussed by the authors in the “Discussion” section.

Nonetheless, one critical point that merits additional attention is the composition of the study group. All twenty-eight samples were obtained from patients already diagnosed with a GCT or another gonadal tumor. To truly assess the diagnostic value of HHGM, the sample panel should also include tissues from patients with diagnoses other than tumors. This would allow for a more robust evaluation of the technique’s ability to distinguish tumor from non-tumor cases across a broader clinical spectrum.

Furthermore, while the conclusion and abstract rightly emphasize the potential of HHGM, they should explicitly state that the accuracy figures are derived from a very limited number of samples. This disclaimer would help temper expectations and reinforce the preliminary nature of the findings.
